# The Key Role of Wettability and Boundary Layer in Dissolution Rate Test

**DOI:** 10.3390/pharmaceutics16101335

**Published:** 2024-10-18

**Authors:** Alice Biasin, Federico Pribac, Erica Franceschinis, Angelo Cortesi, Lucia Grassi, Dario Voinovich, Italo Colombo, Gabriele Grassi, Gesmi Milcovich, Mario Grassi, Michela Abrami

**Affiliations:** 1Department of Engineering and Architecture, University of Trieste, Via Valerio 6/A, I-34127 Trieste, Italy; alice.biasin@phd.units.it (A.B.); federico.pribac@studenti.units.it (F.P.); angelo.cortesi@dia.units.it (A.C.); lucia.grassi@studenti.units.it (L.G.); italo.colombo@protonmail.com (I.C.); michela.abrami@dia.units.it (M.A.); 2Department of Pharmaceutical and Pharmacological Sciences, University of Padova, Via Marzolo 5, I-35131 Padova, Italy; erica.franceschinis@unipd.it; 3Department of Chemical and Pharmaceutical Sciences, University of Trieste, Via Giorgeri 1, I-34127 Trieste, Italy; vojnovic@units.it; 4Clinical Department of Medical, Surgical and Health Sciences, Cattinara University Hospital, Trieste University, Strada di Fiume 447, I-34149 Trieste, Italy; ggrassi@units.it; 5Department of Biological, Chemical and Pharmaceutical Sciences and Technologies, University of Palermo, I-90128 Palermo, Italy; 6Department of Life Sciences, University of Modena and Reggio Emilia, I-41125 Modena, Italy

**Keywords:** DRT, dissolution, particles, wettability, hydrodynamics, mathematical modelling

## Abstract

Background/Objectives: The present work proposes a mathematical model able to describe the dissolution of poly-disperse drug spherical particles in a solution (Dissolution Rate Test—DRT). DRT is a pivotal test performed in the pharmaceutical field to qualitatively assess drug bioavailability. Methods: The proposed mathematical model relies on the key hallmarks of DRT, such as particle size distribution, solubility, wettability, hydrodynamic conditions in the dissolving liquid of finite dimensions, and possible re-crystallization during the dissolution process. The spherical shape of the drug particles was the only cue simplification applied. Two model drugs were considered to check model robustness: theophylline (both soluble and wettable) and praziquantel (both poorly soluble and wettable). Results: The DRT data analysis within the proposed model allows us to understand that for theophylline, the main resistance to dissolution is due to the boundary layer surrounding drug particles, whereas wettability plays a negligible role. Conversely, the effect of low wettability cannot be neglected for praziquantel. These results are validated by the determination of drug wettability performed while measuring the solid–liquid contact angle on four liquids with decreasing polarities. Moreover, the percentage of drug polarity was determined. Conclusions: The proposed mathematical model confirms the importance of the different physical phenomena leading the dissolution of poly-disperse solid drug particles in a solution. Although a comprehensive mathematical model was proposed and applied, the DRT data of theophylline and praziquantel was successfully fitted by means of just two fitting parameters.

## 1. Introduction

DRT (Dissolution Rate Test) is an essential test, widely used in the pharmaceutical field, that involves the dissolution of different solid poly-disperse drug particles within a liquid environment, mainly water or a physiological fluid [1,2]. This is a key drug hallmark, which is strictly connected to its main properties, such as drug solubility, wettability, and particles’ shape and related size distribution. Considering that bioavailability depends on both drug permeability through the cell membrane and drug dissolution properties in physiological fluids [3], DRT can represent a qualitative approach to assess drug bioavailability. Indeed, the drug is absorbed to an extent and rate, becoming available on the site of drug action [3,4]. This is of pivotal importance: about 40% of the drugs currently on the market and 70–90% of new chemical entities are characterized by low dissolution kinetics due to their poor water solubility [4,5,6,7,8]. Therefore, DRT, correlating with the in vivo drug dissolution behavior, represents a key tool in the effective development of pharmaceutical products. For these reasons, drug dissolution profiles and DRT have drawn the attention of many researchers. Hixson and Crowell pioneered the core subject back in 1931 [9,10,11], considering for the first time the effect of surface reduction following the dissolution of spherical particles and established the cubic law. Later on, Niebergall and co-workers [12] observed a deviation from the cubic law and proposed an improvement, assuming that the thickness of the diffusion layer surrounding the dissolving particle was proportional to the square root of the mean particle diameter. The elegant approach of Pedersen and co-workers extended the mathematical modelling of dissolution to poly-disperse spherical particles [13,14,15,16]. Remarkably, this model reduces to the Hixson–Crowell model in monodisperse system models. Other authors stated that the overall dissolution process can be affected by the occurrence of a surface reaction [17] between solute and solvent molecules or by limited solid surface wettability [18]. Regardless of the mechanisms, the final result is a time-dependent drug concentration at the solid–liquid interface that is lower than the drug solubility in the solvent. A similar phenomenon pertains to metastable solids undergoing a phase change (amorphous–crystalline or polymorphic transformation) during the dissolution process, resulting in a time-dependent solid drug solubility [19,20]. Interestingly, the possible drug degradation in the bulk fluid after dissolution was also considered [21]. Of course, researchers focused on the effect of particles shape on dissolution as well [22]. While Hirai and co-workers did not explicitly consider the shape of the particles, they proposed a law that can describe the dissolution surface in a time-dependent fashion [23]. Abrami et al. dealt with spherical, cylindrical, and parallelepiped particles [20]. Hsu and Wu [24] considered sphere-, cylinder-, bi-cone-, cone-, and inverse-cone-shaped particles, whereas Yuan and co-workers focused on the dissolution of irregularly shaped particles [25]. Therefore, particle shape is connected with two additional core dissolution features, i.e., drug concentration profile in the boundary layer (BL) surrounding the solid surface and BL thickness. Indeed, it can be demonstrated that drug concentration profile is not linear, as originally assumed [26,27,28,29], unless dissolution occurs from a flat surface. Moreover, BL thickness depends both on dissolution medium hydrodynamic conditions and particle dimension, as documented by D’Arcy and Persoons [30,31]. Recently, Abrami and co-workers gathered most of the parameters correlated with dissolution in an effective mathematical model [32]. They focused mainly on particle shape, specifically on the local surface curvature radius, describing the dissolution both from concave and convex surfaces. Hence, the dissolution of solid particles of any shape can be considered.

Therefore, this work aims to elucidate a tricky, still-pivotal issue: the combined effect exerted on DRT kinetics both by the boundary layer surrounding each drug particle and the drug wettability. At this purpose, the DRT profiles of two different model drugs (theophylline, TPH—good solubility and wettability—and praziquantel, PRQ—poor solubility and wettability) were studied. Indeed, a mathematical model was proposed, with the core phenomena involved as the dissolution of polydisperse particles. Herein, it is hypothesized that the model relies on the immediate attainment of pseudo-stationary conditions, which concern drug mass transport inside the boundary layer. Despite the general complexity of the developed mathematical model, its numerical solution and its data fitting to experimental data were realized within a Microsoft Excel sheet as a user-defined function. This proposes the model as very user-friendly and, thus, targeted to a broad audience plethora, as its application is suitable also for researchers not generally used to dealing with mathematical models. Moreover, the simple solution strategy proposed fosters an approach particularly suited for an industrial environment, too. Herein, rapid and precise answers are usually needed. Hence, it is very important to understand the relative importance of different phenomena in order to improve the performance of drug delivery systems.

## 2. Materials and Methods

### 2.1. Drugs

The first model drug considered was theophylline (TPH), a bronco-dilatator indicated for the treatment of asthma, bronchospasm, and Chronic Obstruction Pulmonary Diseases (COPD) (Carlo Erba, Milano, Italy; C_7_H_8_N_4_O_2_·H_2_O, Mw = 198.2; essentially a neutral compound characterized by water solubility of 12,495 μg/mL at 37 °C and maximum UV absorbance at wavelength 272 nm [33]). The second model drug was praziquantel (PRQ), used to treat the infections of different parasites, such as schistosomiasis (kind gift by Fatro S.p.A., Bologna, Italy. C_19_H_24_N_2_O_2_, Mw = 312.4; water solubility 180 μg/mL at 37 °C and maximum UV absorbance at wavelength 262.6 nm [34,35]).

### 2.2. Wettability

Drugs wettability was evaluated at 25 °C by measuring the liquid–solid contact angles of four liquids (deionized water (H_2_O), formamide (CH_3_NO), dimethyl sulfoxide ((CH_3_)_2_SO), and diiodomethane (CH_2_I_2_), (Sigma-Aldrich, Milano, Italy)) on compacted drug powder measured on a DSA 10 tensiometer (Krüss, Hamburg, Germany) connected to the DSA 4 software (Krüss, Germany). For this purpose, ca. 200 mg of each drug were compressed using a single punch tablet machine (Cosalt, Officina CO.STA, Gorgo al Monticano, Italy) equipped with a 10 mm flat punch. The liquid drop spreading onto the tablets’ surface was recorded using the fast digital camera of the tensiometer. The videos were processed by means of the instrument software, and the contact angle (θ) was evaluated by applying the tangent method (T-1) to the image where the base diameter did not increase [36]. Each measurement was performed in triplicate and is reported in Table 1 as mean value ± standard deviation.

Based on the contact angle measurements, drug wettability was estimated according to the spreading coefficient *S*_C_ defined by the following [37]:(1)$$S_{C}=\gamma_{\mathrm{lv}}\left(\cos(\theta )-\;1\right)$$
where θ is the solid–liquid contact angle while γ_lv_ is the liquid–vapor surface energy that, for the four liquids considered, is reported in Table 2, as well as the polar and dispersion components [18]. The data reported in Table 1 and Table 2 allow us to evaluate the polarity of the model drug considered, following the Wu approach [38]. According to this approach, the solid–vapor surface energy (γ_sv_) is the sum of a polar (γ_sv_^p^) and a dispersion component (γ_sv_^d^). The two γ_sv_ components can be evaluated by the simultaneous solution of the following system of nonlinear equations:(2)$$\left\{\begin{array}{c} \gamma_{l1v}\left(1+\cos\theta_{1}\right)=\frac{4\gamma_{\mathrm{sv}}^{d}\gamma_{l1v}^{d}}{\gamma_{\mathrm{sv}}^{d}+\gamma_{l1v}^{d}}+\frac{4\gamma_{\mathrm{sv}}^{p}\gamma_{l1v}^{p}}{\gamma_{\mathrm{sv}}^{p}+\gamma_{l1v}^{p}}\\ \gamma_{l2v}\left(1+\cos\theta_{2}\right)=\frac{4\gamma_{\mathrm{sv}}^{d}\gamma_{l2v}^{d}}{\gamma_{\mathrm{sv}}^{d}+\gamma_{l2v}^{d}\;}+\frac{4\gamma_{\mathrm{sv}}^{p}\gamma_{l2v}^{p}}{\gamma_{\mathrm{sv}}^{p}+\gamma_{l2v}^{p}} \end{array}\right.$$
where θ_1_ and θ_2_ are the contact angles referring to a polar (water) and a non-polar (diiodomethane) fluid, respectively (Table 1), γ_1lv_ and γ_2lv_ are the liquid–vapor surface energies referring to the two liquids, and γ_1lv_^p^, γ_2lv_^p^, γ_1lv_^d^, and γ_2lv_^d^ are, respectively, their polar and dispersion components reported in Table 2. The solution of the equation system is reported in Table 3. In order to confirm the Wu approach, γ_sv_ was evaluated according to the state equation approach proposed by Kwok, Neumann, and Li [39,40] as per all the four contact angles shown in Table 1. This check supported the Wu approach, as we detected γ_sv_ (mJ/m^2^) = 61 (TPH) and γ_sv_ (mJ/m^2^) = 47 (PRQ).

### 2.3. Particle Size

The particle size distribution (PSD) of TPH and PRQ powders was evaluated by Dynamic Laser Light Scattering (Mastersizer Hydro 2000, Malvern Instruments, Malvern, UK) using as a dispersant liquid silicone oil (cyclomethicone K4, ACEF, Fiorenzuola d’Arda, Italy) for theophylline and deionized water plus 1% (*w*/*w*) polysorbate 80 (ACEF, Italy) for praziquantel.

Powders were dispersed in a small amount of dispersant. The suspensions were mixed by magnetic stirring and then added to the instrument’s dispersion unit, containing about 200 mL of dispersing liquid, until the quenching reached a value between 10% and 20%. The analysis was performed in triplicate, using a dispersion unit controller set to 1800 rpm.

Particle size distributions were calculated according to Mie theory [41] by means of the following refractive index values: 1.330 for silicone oil, 1.360 for water, and 1.700 for theophylline and praziquantel. For mathematical modeling purposes, we hypothesized that particles were spherical. The cumulative PSDs referring to the TPH and PRQ are reported in Figure 1.

In the light of the mathematical model proposed to study DRT, the Weibull distribution [42] was fitted to the experimental PSD referring to the two model drugs considered:(3)$$W\%=100\times (1-e^{\left(-{\left(2\frac{R-R_{min}}{\phi }\right)}^{\varphi }\right)})$$

The values of the fitting parameters jointly with the average radius (*R*_A_) are reported in Table 4.

### 2.4. DRT Test

A pre-determined amount of drug (3 mg TPH; 27 mg PRQ) was dispersed in 150 mL of distilled-degassed water contained in a proper thermostatic glass vessel (37 °C). As both PRQ solubility and molar extinction (ε) are very low, it was necessary to consider a high amount of this drug to be able to check and record the beginning part of the DRT process. On the other hand, TPH did not show this problem, and hence a smaller amount could be considered in the DRT experiments. A magnetic stirrer, lying on the vessel bottom and rotating at 370 rpm, ensured fluid mixing. Drug concentration in the dissolution environment was measured by an optical fiber (HELLMA, Milano, Italy) ending with a probe characterized by an optical path of 5 mm (TPH) or 10 mm (PRQ). With this setup, absorbance was always ≤1. Probe distance from the stirrer was about 5 cm to prevent bubble adhesion on it. The optical fiber was connected to a UV spectrophotometer (ZEISS, MCS 600, Oberkochen, Germany) to record absorbance. In order to avoid the scattering effect of solid particles, the absorbance determined at each drug wavelength (272 nm TPH; 262.6 nm PRQ) was deprived by the absorbance recorded at 500 nm (i.e., very far from the drug wavelength). Indeed, the effect of solid particle scattering is the same whatever the wavelength. Absorbance (ABS) recording started just after drug dispersion in the dissolving medium. Molar extinction (ε) of TPH and PRQ were 12,117 (M^−1^ × cm^−1^) and 342.3 (M^−1^ × cm^−1^), respectively (see Appendix A). All tests were performed in triplicate.

## 3. Mathematical Modelling

The dissolution phenomenon of a solid in a liquid environment is associated with four steps that can be seen as energetic barriers, hindering the dissolution process [20]. These refer to (1) contact of the solvent with the solid surface (wetting; Δ*E*_w_), (2) breakdown of intermolecular bonds in the solid phase (fusion; Δ*E*_f_), (3) molecules’ transfer from the solid phase to the solid–liquid interface (solvation; Δ*E*_s_), (4) diffusion of the solvated molecules through the unstirred boundary layer (BL) surrounding the solid surface (diffusion; Δ*E*_d_) (see Figure 2). These steps represent the total resistance for the drug molecules’ movement from the solid phase to the solution one (dissolution). Moreover, the first three steps are connected to the surface resistance (*R*_m_) for drug dissolution, while the last one (*R*_d_ = δ/*D*) represents the drug resistance to cross the BL of thickness (d) and drug diffusivity *D* (see Figure 2). Indeed, *R*_d_ is connected to the hydrodynamic conditions of the liquid environment. As the present work aims to model the dissolution from different poly-disperse spherical drug particles, it is necessary to consider the contribution to dissolution due to each one (ith class) of the *N* dimensional classes, into where the continuous particle size distribution can be split. Indeed, dissolution kinetics depends on particle radius. Thus, from now on, we will focus on the development of our model, drawing attention to the ith class.

Fick’s second law represents the starting point to blend the four dissolution steps. Herein, it is hypothesized that pseudo-stationary conditions hold within the BL, so that Fick’s second law reads as follows [32]:(4)$$\nabla \left(D\nabla C_{i}\right)=0$$
where *C*_i_ is the position-dependent drug concentration inside the BL of the ith class. Assuming that mass transport essentially occurs in the radial direction (ξ^i^), Equation (4) must be solved according to the following boundary conditions:(5)$${\left.\left(D\nabla C_{i}\cdot n_{i}\right)\right|}_{\xi^{i}=\xi_{\min}^{i}}=-k_{m}(C_{S}-C(\xi_{\min}^{i}))$$
(6)$$C\left(\xi_{\max}^{i}\right)=C_{b}$$
where $\xi_{\max}^{i}$ and $\xi_{\min}^{i}$ are the radii defining the BL thickness δ_i_ (see Figure 2), *C*_s_ is the drug solubility in the dissolution medium, ***n*_i_** is the normal vector to the particle surface, *C*_b_ is the time-dependent drug concentration in the dissolution medium, and *k*_m_ is the mass transfer coefficient related to the first three steps of the dissolution process and mainly depending on the surface wettability.

Equation (5) requires that the drug flux leaving the solid surface is driven by both *k*_m_ and the difference between *C*_s_ and the drug concentration at the solid–liquid interface (ξ^i^ = $\xi_{\min}^{i}$). Equation (6) states that the drug concentration at the BL-dissolution medium (ξ^i^ = $\xi_{\max}^{i}$) is equal to the bulk concentration *C*_b_.

Although *k*_m_ depends on the surface curvature radius ($\xi_{\min}^{i}$), thus it should be class-dependent ($k_{m}^{i}$), this dependence is so small that it can be neglected and *k*_m_ can be assumed constant for every *N* class [32]. Conversely, the drug resistance, $R_{d}^{i}$ = δ_i_/*D*, due to the drug crossing the BL of thickness δ_i_ and drug diffusion coefficient *D* (step 4), depends on many physical parameters, such as the particle curvature radius ($\xi_{\min}^{i}$). D’Arcy and Persoons nicely modelled this dependence in terms of the hydrodynamics mass transfer coefficient $k_{d}^{i}$(=1/$R_{d}^{i}$; also referred to as intrinsic dissolution constant) [30,31]:(7)$$k_{d}^{i}=\frac{D}{2\xi_{\min}^{i}}\;(2+0.6\sqrt{∆U\frac{2\xi_{\min}^{i}}{\nu_{f}}}{\left(\frac{\nu_{f}}{D}\right)}^{\frac{1}{3}})\;\;∆U\;=\alpha \frac{\left(\rho_{s}-\rho_{f}\right)}{\rho_{f}}g\frac{{\left(2\xi_{\min}^{i}\right)}^{2}}{18\nu_{f}}$$
where ρ_s_ and ρ_f_ are, respectively, the solid drug and the fluid density, ν_f_ is the fluid kinematic viscosity, *g* is the gravity acceleration, Δ*U* is the absolute relative velocity between particles and fluid, and α is an adjustable parameter (≥0), which was set to 1 on the original D’Arcy and Persoons model.

Equation (4)’s analytical solution, in the light of boundary conditions (Equations (5) and (6)), reads as follows:(8)$$C_{i}\left(\xi^{i}\right)=C_{b}+\left(C_{s}-C_{b}\right)\frac{\frac{k_{m}}{k_{d}^{i}}{\left(\xi_{\min}^{i}\right)}^{2}\left(\frac{k_{d}^{i}}{D}\left(\frac{\xi_{\min}^{i}}{\xi^{i}}-1\right)+\frac{1}{\xi^{i}}\right)}{\xi_{\min}^{i}(1+\frac{k_{m}}{k_{d}^{i}})+\frac{D}{\;k_{d}^{i}}}\;$$

Equation (8) clearly shows that drug concentration inside BL is not linearly dependent on the radial position ξ^i^. Based on Equation (8) it is possible to determine the drug concentration at the solid drug–BL interface ($C_{0}^{i}$ = *C*($\xi_{\min}^{i}$)):(9)$$C_{0}^{i}=C_{b}+\left(C_{s}-C_{b}\right)\frac{\xi_{\min}^{i}}{\frac{k_{d}^{i}}{k_{m}}\xi_{max}^{i}+\xi_{\min}^{i}}$$

Equation (9) states that $C_{0}^{i}$ is always less than *C*_s_ and it equates *C*_s_ only when *C*_b_ = *C*_s_, i.e., after a very long time and when the solid drug amount is sufficient to obtain *C*_s_ in the dissolution environment. Clearly, when *k*_m_ is very large (i.e., the mass transfer resistance *R*_m_ = 1/*k*_m_ is vanishing), $C_{0}^{i}$ immediately equates *C*_s_. On the contrary, when surface wettability is very poor, *k*_m_ is very small (i.e., the mass transfer resistance *R*_m_ = 1/*k*_m_ is very big) and $C_{0}^{i}$ is very close to *C*_b_ so that dissolution kinetics will be very slow.

As it is quite common that, during the dissolution process, the drug undergoes a phase transformation (polymorphic or amorphous–crystalline), drug solubility can reduce over time. This phenomenon is usually described by a first order reaction [43] occurring at the solid–liquid interface and leading to the following expression for the *C*_s_ temporal reduction:(10)$$C_{s}=C_{\mathrm{sf}}+(C_{s-\mathrm{in}}-C_{\mathrm{sf}})e^{\left(-k_{r}t\right)}$$
where *C*_sf_ and *C*_s−in_ are, respectively, the final and initial values of solubility, while *k*_r_ is the recrystallization constant and *t* is time. Indeed, Equation (10) refers to the dissolution step 2, as solubility is directly connected with the crystal network breakdown attitude, quantified by its melting temperature and enthalpy [44].

In order to evaluate the particles’ radii ($\xi_{\min}^{i}$) time decrease, it is necessary to consider *N* ordinary differential equations (one for each class) referring to the particles mass (*M*_i_) reduction:(11)$$\frac{dM_{i}}{dt}=\frac{d}{dt}\left(\rho_{s}\frac{4}{3}\pi {\left(\xi_{\min}^{i}\right)}^{3}\right)=4\pi {\left(\xi_{\max}^{i}\right)}^{2}{\left.\left(D\frac{\partial C_{i}}{\partial \xi^{i}}\right)\right|}_{\xi^{i}=\xi_{\max}^{i}}$$

Equation (11) states that the time reduction of the ith particle mass equates the mass amount leaving the particle through the surface located at the end of the BL (ξ^i^ = $\xi_{\max}^{i}$). As Equation (8) allows us to evaluate the partial derivative appearing in Equation (11), after some algebraic manipulations, Equation (11) can be re-written in a simpler and more straightforward form:(12)$$\frac{d\xi_{\min}^{i}}{\mathrm{dt}}=\frac{K_{i}}{\rho_{s}}\left(C_{s}-C_{b}\right)\;\;\;\;K_{i}=\frac{\xi_{\min}^{i}/\xi_{\max}^{i}}{(\frac{1}{k_{d}^{i}}+\frac{1}{k_{m}}(\frac{\xi_{\max}^{i}}{\xi_{\min}^{i}}))}$$
where *K*_i_ is the overall mass transport coefficient referred to both $k_{d}^{i}$ and *k*_m_. Its inverse represents the global mass transport resistance (*R*_i_), sum of *R*_m_ and $R_{d}^{i}$. Obviously, Equation (12) works only when *C*_b_ < *C*_s_, i.e., when particle dissolution can take place. When the above mentioned condition does not take place, ξ^i^ would be constant, as its time derivative would be zero (second right term of Equation (12)). In this case, part of the already dissolved drug inside the dissolution medium would precipitate, and this phenomenon can be modelled as per the following first order equation:(13)$$\frac{dM_{c}}{dt}=k_{\mathrm{rb}}V(C_{s}(t)-C_{b}(t))$$
where *M*_c_ is the amount of re-crystallized drug (*M*_c_ = 0 at the beginning of the dissolution process), *V* is the dissolution volume, and *k*_rb_ is the bulk re-crystallization constant, which is usually assumed to be equal to *k*_r_ [20] (see Equation (10)). In order to balance unknowns and equations, it is possible to refer to an overall mass balance, where, at any time, the initial solid mass (*M*_0_) must be equal to the sum of the undissolved drug mass, the solubilized drug present in the bulk solution, and *M*_c_:(14)$$M_{0}=\sum_{i=1}^{i=N}\rho_{s}N_{\mathrm{pi}}V_{\mathrm{pi}}+C_{b}\left(t\right)V+M_{c}(t)\;\mathrm{or}\;C_{b}\left(t\right)=\frac{M_{0}-\sum_{i=1}^{i=N}\rho_{s}N_{\mathrm{pi}}V_{\mathrm{pi}}-M_{c}\left(t\right)}{V}$$
where *N*_pi_ indicates the number of particles belonging to the ith class and, thus, sharing the same radius at the beginning of the dissolution process ($\xi_{\min-0}^{i}$), which can be deduced from the Weibull equation (Equation (3)) characterizing the particle size distribution:(15)$$N_{\mathrm{pi}}=V_{0}\frac{\left(V_{i}-V_{i-1}\right)}{\frac{4}{3}\pi {\left(\xi_{\min-0}^{i}\right)}^{3}}\;\;\;V_{\mathrm{pi}}=\frac{4}{3}\pi {\left(\xi_{\min}^{i}\right)}^{3}$$
where *V*_0_ is the particles volume (=*M*_0_/ρ_s_) while *V*_i_ and *V*_i−1_ are defined by the following:(16)$$V_{i}=1-e^{\left(-{\left(2\frac{\xi_{\min}^{i}-R_{\min}}{\phi }\right)}^{\varphi }\right)}\;\;\;V_{i-1}=1-e^{\left(-{\left(2\frac{\xi_{\min}^{i-1}-R_{\min}}{\phi }\right)}^{\varphi }\right)}$$
where *R*_min_ represents the smallest particle’s radius at the beginning of the dissolution. The numerical solution of model equations (see Appendix B for details) allows us to determine the time variation of the bulk drug concentration (*C*_b_) as well as the drug concentration inside the BL for each of the *N* particles classes.

## 4. Results and Discussion

### 4.1. Microscopic Results

Before checking the model on the experimental DRT data, it is interesting to look at the microscopic information that it provides: the evolution of the drug concentration profile inside the BL during the dissolution process. Such a feature has never been completely handled by other mathematical models. Therefore, we assume that drug recrystallization does not occur—this would just entangle the analysis, with no specific, useful tool related to the physical considerations investigated herein. Furthermore, a monodisperse particle size distribution is considered (thus, superscript/subscript “i” will be cut in this section) and usual values are fixed for the parameters leading the dissolution process (see caption to Figure 3). Equation (8) allows us to draw the variation of the drug concentration profile inside the BL during the dissolution process. Clearly, *C*_b_ (or its dimensionless expression *C*_b_^+^ = *C*_b_/*C*_s_) is evaluated through the mass balance (Equation (14)), where, based on previous hypotheses, only one class is considered and *M*_c_ = 0. Moreover, Equation (9) allows us to evaluate the drug concentration at the solid–liquid interface, i.e., in ξ = ξ_min_. In order to broaden the model applications, dimensionless concentration (*C*^+^ = *C*/*C*_s_) and radial position (ξ^+^ = ξ/ξ_min−0_; ξ_min−0_ is the initial particle radius) have been set. Figure 3A considers a solid drug that does not show wettability issues, i.e., the *k*_m_/*k*_d_ ratio is high (between 10 and 10^2^) or, equivalently, the surface resistance *R*_m_ (=1/*k*_m_) is small, compared to the hydrodynamic one *R*_d_ (=1/*k*_d_). Based on the *k*_d_ variation with radius (Equation (7)), a range of *k*_m_/*k*_d_ is applied, instead of only one value. The red lines in Figure 3A represent the evolution of the drug concentration profile (Equation (8)) during the dissolution process.

In detail, over Figure 3A, the first red line on the right refers to the drug concentration profile at the beginning (*t* = 0) of the dissolution process. Vertical dashed and dotted lines indicate the dimensionless BL thickness (δ^+^ = *D* × *k*_d_/ξ_min-0_). Thus, the lowermost part of each profile shows the end of the BL (ξ = ξ_max_), while the uppermost part indicates the beginning of the BL defined by the solid–liquid interface (ξ = ξ_min_). As the dissolution proceeds, the drug profile concentration moves on the left, as well as particle radius (ξ_min_) reducing up to the end of the dissolution (particle disappearing). Furthermore, δ^+^ reduces as dissolution proceeds, according to the distance separating dashed or dotted vertical lines. Considering that wettability issues do not occur, the dimensionless drug concentration at the solid–liquid interface (dashed green line, *C^+^*(ξ^+^_min_) = *C*/*C*_s_) is equal to one, except at the end of the dissolution, when it almost merges to *M*_0_/(*V* × *C*_s_), set to 0.625 in our simulations. Thus, drug concentration at the solid–liquid interface is equivalent to drug solubility in the dissolution environment *C*_s_. The blue dashed line represents the trend of the dimensionless drug concentration at the end of the BL, which, according to Equation (6), equals the dimensionless bulk drug concentration *C*_b_^+^. Therefore, *C*_b_^+^ starts from zero and terminates at the final value of 0.625 (=*M*_0_/(*V* × *C*_s_)). When solid wettability decreases (Figure 3B; 0.2 < (*k*_m_/*k*_d_) < 3), drug concentration at the solid–liquid interface (ξ = ξ_min_) is never equal to one (see dashed green line), and thus the concentration gradient across the BL is lower, leading to slower dissolution kinetics. Finally, when wettability issues are relevant (Figure 3C), the dimensionless drug concentration at the solid–liquid interface (dashed green line, *C^+^*(ξ^+^_min_)) increases with a monotonic trend from zero up to the final value of 0.625. Therefore, the concentration gradient across BL is fading and the kinetics of the dissolution process are strongly decreased. Overall, the analysis of Figure 3A–C reveals some important theoretical features of the proposed model, i.e., that the evolution of the BL thickness δ^+^ is not affected by solid wettability. On the other hand, it influences the drug concentration profile inside BL, hence the drug concentration at the solid–liquid interface (ξ = ξ_min_).

Remarkably, the *C*_b_^+^ evolution (blue dashed lines) is identical for the three wettability conditions examined in Figure 3A–C. This is due to the *C*_b_^+^ evolution, which is referred to the reduction of the particle radius and not to time increase. Indeed, the same radius reduction requires an increasing time when drug wettability reduces. Thus, one of the advantages of the proposed model is to provide a kind of analytical solution explaining the evolution of the drug concentration inside BL when a particle dissolves based on a monodisperse distribution (Equations (8) and (9)). Indeed, in order to connect radius reduction to elapsing time, it is required to numerically solve the whole model.

### 4.2. Macroscopic Results: Data Fitting

#### 4.2.1. Theophylline

Figure 4 shows the model’s best fitting (solid line) to experimental DRT data (symbols) referring to TPH. In detail, TPH monohydrate is stable in the aqueous dissolution environment. Indeed, it does not undergo re-crystallization upon dissolution. Consequently, the profile concentration does not show the usual slope reduction at the beginning, induced by the solubility reduction [18]. The data fitting is very accurate, and the values of the two model fitting parameters read α = 27 and *k*_m_ = 0.38 m/s. The fact that α is >1 involves a “disengagement” from the original D’Arcy and Persoons equation for the $k_{d}^{i}$ evaluation (Equation (7)). Thus, we conclude that the relative velocity between particles and fluid is greater than that proposed by D’Arcy and Persoons. This is not surprising as their approach, although very useful and smart, represents an approximation of what really occurs between particles and a dissolving fluid. Moreover, it is worth mentioning that their theory refers to spherical particles, while TPH particles appear as solid bodies, i.e., parallelepipeds [20]. The comparison between *k*_m_, $k_{d}^{i}$ (Equation (7)) and *K*_i_ (Equation (12)), shown in Figure 5, reveals that *k*_m_ is >>$k_{d}^{i}$; thus, the overall mass transport coefficient *K*_i_ almost matches with $k_{d}^{i}$. Essentially, the most important resistance to TPH dissolution is represented by the existence of the BL surrounding each particle, whose thickness is strictly connected to the hydrodynamic conditions of the dissolution medium. Thus, the analysis of DRT data by means of the proposed model reveals that TPH is not affected by wettability problems.

#### 4.2.2. Praziquantel

Figure 6 reports the model best fitting (solid line) to experimental DRT data (symbols) referring to PRQ. In this situation, no re-crystallization occurs too, as the aqueous dissolution environment did not induce any PRQ structure transformation. Therefore, the profile concentration did not show the usual slope reduction at the beginning due to a solubility reduction [18]. The quality of data fitting is very accurate and the values of the two model fitting parameters read α = 1 and *k*_m_ = 0.001 m/s. Remarkably, for PRQ, α turns out to be equal to one, as assumed by D’Arcy and Persoons (Equation (7)). As the shape of PRQ particles is not too different from that of TPH (PRQ particles resemble needles [34]), we hypothesize that the difference compared to TPH (α = 27) is not related to shape. Vice versa, as TPH particles are definitely bigger than the PRQ ones (see Figure 1), α should be mainly affected by particles’ dimension.

In order to test this hypothesis, further investigation on the DRT of particles characterized by different shapes and different size distributions is required. The comparison between *k*_m_, $k_{d}^{i}$ (Equation (7)) and *K*_i_ (Equation (12)) shown in Figure 7 clearly reveals that *k*_m_ is comparable with $k_{d}^{i}$ (although bigger) and that the overall mass transport coefficient *K*_i_ depends on both *k*_m_ and $k_{d}^{i}$. The non-negligible contribution to *K*_i_ due to *k*_m_ demonstrates that $k_{d}^{i}$ > *K*_i_, as clearly shown in Figure 7. Thus, the mass transfer resistance connected to the PRQ dissolution depends on both the surface (*R*_m_ = 1/*k*_m_ = 10^3^ s/m) and the hydrodynamic ($R_{d}^{i}$ = 1/$k_{d}^{i}$ ≈ 10^3^–10^4^ s/m) resistances. Therefore, PRQ’s slow dissolution kinetics are due to the combination of low solubility and moderate wettability. Indeed, the analysis of DRT data by means of the proposed model allows us to determine and quantify the effect on DRT of some key drug features, such as solubility, wettability, and particles’ size distribution. This task would be not easily accomplished otherwise.

#### 4.2.3. Comparison Between Drugs

The analysis of DRT data by means of the proposed mathematical model is confirmed by the results of drugs surface characterization, summarized in Figure 8. Indeed, for PRQ, drug wettability decreases with fluid polarity (solid line), as can be observed from the progressive decrease in the spreading coefficient (*S*_C_, Equation (1)) with fluid polarity. Remarkably, *S*_c_ variation between water and diiodomethane is considerable, (around 50 mJ/m^2^). On the other hand, for TPH, wettability reaches a maximum when fluid polarity is between 20 and 30% (*S*_c_ zeroes), while it decreases for higher and lower fluid polarities (dashed line in Figure 8). In this case, *S*_c_ variation is about 25 mJ/m^2^, i.e., 50% of the that of PRQ. These considerations match with the TPH polarity feature, which is ≈51% (an almost amphiphilic behavior; dashed vertical line in Figure 8), while PRQ’s polarity is ≈25% (a typical a-polar behavior; solid vertical line in Figure 8), i.e., one half of that of TPH. Thus, the lower PRQ polarity compared to that of TPH is perfectly compatible with the lower *k*_m_ value associated with PRQ with reference to that of TPH.

However, the different dissolution behavior of TPH and PRQ is also due to different $k_{d}^{i}$ (about one order of magnitude bigger in the TPH case, see Figure 5 and Figure 7). This leads to a smaller BL average thickness in the TPH case (≈0.1 μm) with reference to PRQ (≈3.7 μm). The combination of surface wettability and BL thickness makes the resistance to drug dissolution of PRQ about 35 times higher than that of TPH. Bearing in mind that PRQ solubility is about two orders of magnitude lower than that of TPH, PRQ dissolution is remarkably slow compared to that of TPH.

## 5. Conclusions

Despite the complexity of the mathematical model developed, we showed that the DRT data for both model drugs (theophylline and praziquantel) were successfully fitted by means of just two fitting parameters. In detail, the mentioned parameters are *k*_m_, the mass transfer coefficient related to the first three steps of the dissolution process and mainly depending on the surface wettability, and α (see Equation (7)), which refers to the effect on dissolution of the relative velocity between particles and dissolution medium. Moreover, the presented mathematical model proposes as an elegant, user-friendly tool which can be run in a simple, yet effective, Microsoft Excel sheet as a user-defined function. Thus, data fitting was performed using the Solver functionality. Hence, the proposed mathematical model revealed two main outcomes as a result of the data interpretation in terms of micro- and macroscopic aspects. As concerns the microscopic aspects, we were able to determine how the relative importance of solid wettability and boundary layer, represented by the *k*_m_/*k*_d_ ratio, affects the drug concentration profile inside the boundary layer during the entire dissolution process. As this time evolution cannot be experimentally detected, the proposed model behaved as a sort of theoretical microscope and allowed us to understand the missing insights into DRT. On the other hand, as concerns the macroscopic aspects, a comprehensive, robust strategy was proposed regarding the role of drug properties (mainly wettability and solubility), particle size distribution, and dissolution medium hydrodynamics on the DRT kinetics. Data fitting confirmed the importance of the different physical phenomena leading the dissolution of different poly-disperse solid drug particles. In particular, we found a perfect match between model outcomes in terms of *k*_m_ value and drug wettability evaluated through the spreading coefficient and the surface polarity. Hence, the mathematical model showed a reliable hallmark, able to evaluate the relative importance of the most pivotal phenomena involved in the DRT process, the key information required in experimental, industrial, and theoretical applications.

## Figures and Tables

**Figure 1 pharmaceutics-16-01335-f001:**
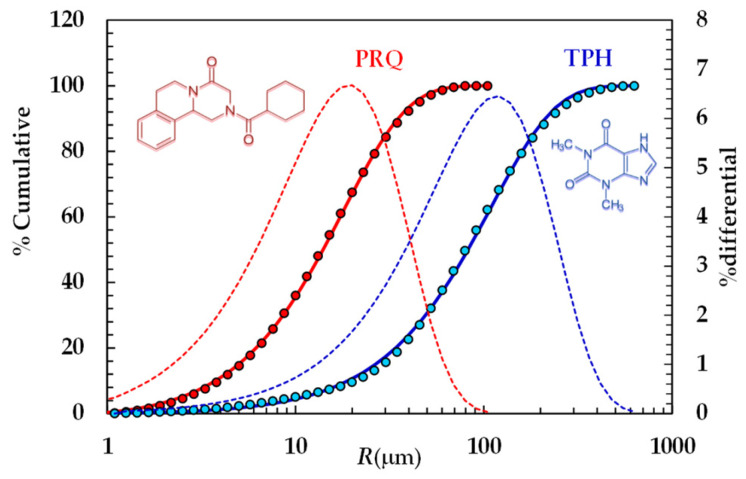
Particle size distribution referring to TPH and PRQ (symbols). Continuous lines indicate the best fitting of the cumulative Weibull distribution (Equation (3)), while dotted lines indicate the differential particle size distribution (right vertical axis).

**Figure 2 pharmaceutics-16-01335-f002:**
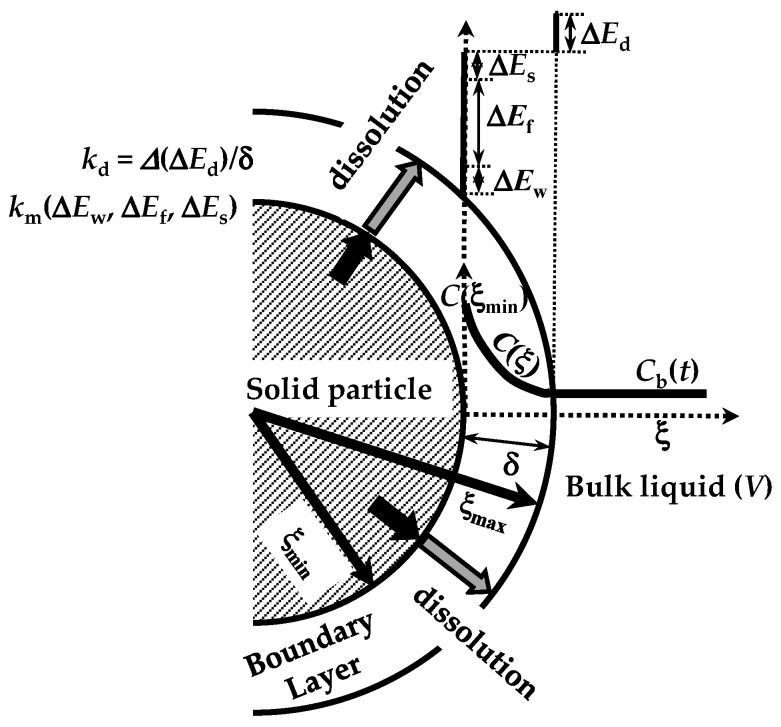
Four energetic barriers hinder the dissolution of a solid drug in a solvent: solid wetting (Δ*E*_w_), breakdown (fusion) of solid molecular bonds (Δ*E*_f_), drug molecules’ solvation (Δ*E*_s_), and drug molecules’ diffusion through the boundary layer surrounding the solid particle (Δ*E*_d_). These energies affect, in different manners, the mass transfer coefficient (*k*_m_) at the solid–liquid interface and the dissolution constant *k*_d_. Notably, due to possible solid surface wetting problems, the drug molecule concentration at the solid–liquid interface (*C*(ξ_min_)) can be lower than drug solubility in the dissolution medium. ξ indicates the radial coordinate while *V* is the dissolution environment volume. Adapted from [28].

**Figure 3 pharmaceutics-16-01335-f003:**
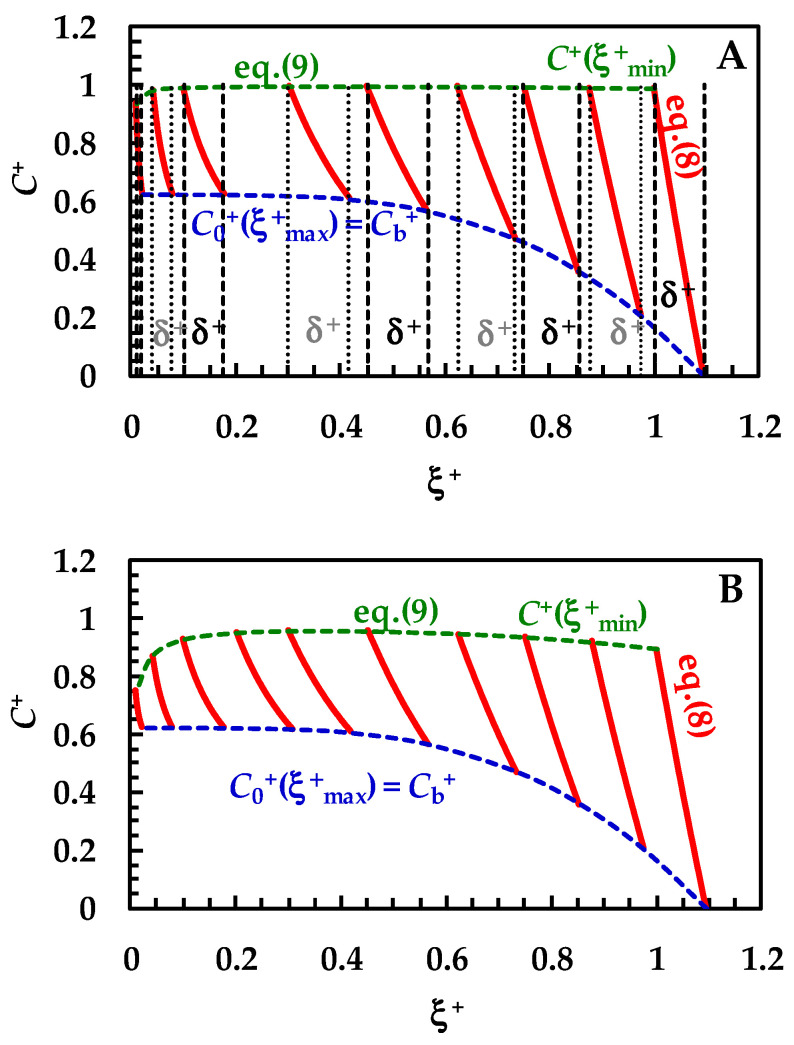
Temporary evolution of the dimensionless drug profile concentration (red lines, Equation (8)) inside BL (ξ^+^ is the dimensionless radial coordinate). *C*_b_^+^ (=*C*_b_/*C*_s_) is the dimensionless drug concentration inside the dissolution medium (dashed blue lines) being *C*_s_ drug solubility (assumed constant in time). Dashed green lines (Equation (8)) indicate the dimensionless drug concentration at the solid/liquid interface (ξ = ξ_min_). Three different ranges for the *k*_m_/*k*_d_ ratio were considered: (**A**) 10 < (*k*_m_/*k*_d_) < 10^2^ (no wettability issues. Vertical dashed and dotted lines indicate dimensionless BL thickness δ^+^), (**B**) 0.2 < (*k*_m_/*k*_d_) < 3 (moderate wettability issues), and (**C**) 10^−3^ < (*k*_m_/*k*_d_) < 10^−2^ (considerable wettability issues). All other parameters are equal and read as follows: ρ_s_ = 1500 kg/m^3^, ρ_f_ = 1000 kg/m^3^, η(Pa s) = 10^−3^, ν_f_(m^2^/s) = 10^−6^, *D*(m^2^/s) = 10^−10^, *C*_inf_/*C*_s_ = 0.625 (*C*_inf_ is the drug concentration reached in the dissolution medium upon complete dissolution of the solid drug particles), *k*_r_ = 0, α = 1, and *g* = 9.81 m/s^2^. These values are typical of small organic drugs such as those considered in this work.

**Figure 4 pharmaceutics-16-01335-f004:**
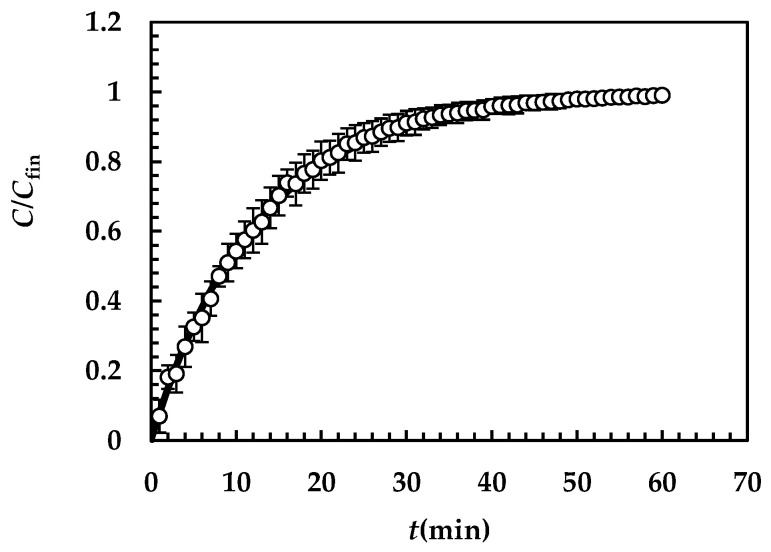
Model’s best fitting (solid line) to experimental DRT data (symbols) referring to TPH (37 °C). Vertical bars indicate data standard error. The physical parameters adopted to perform data fitting read as follows: ρ_s_ = 1490 kg/m^3^, ρ_f_ = 993 kg/m^3^, η(Pa s) = 6.91 × 10^−3^, ν_f_(m^2^/s) = 0.696 × 10^−6^, *D*(m^2^/s) = 8.26 × 10^−10^, and *C*_s_ (kg/m^3^) = 12.49 [20] while *k*_r_ = *k*_rb_ = 0 as monohydrate TPH does not undergo re-crystallization upon dissolution. TPH particle size distribution is described by the Weibull distribution (Equation (3)), whose parameters are those reported in Table 4. Concentration data (*C*) are normalized with respect to the final concentration *C*_fin_ = *M*_0_/*V*.

**Figure 5 pharmaceutics-16-01335-f005:**
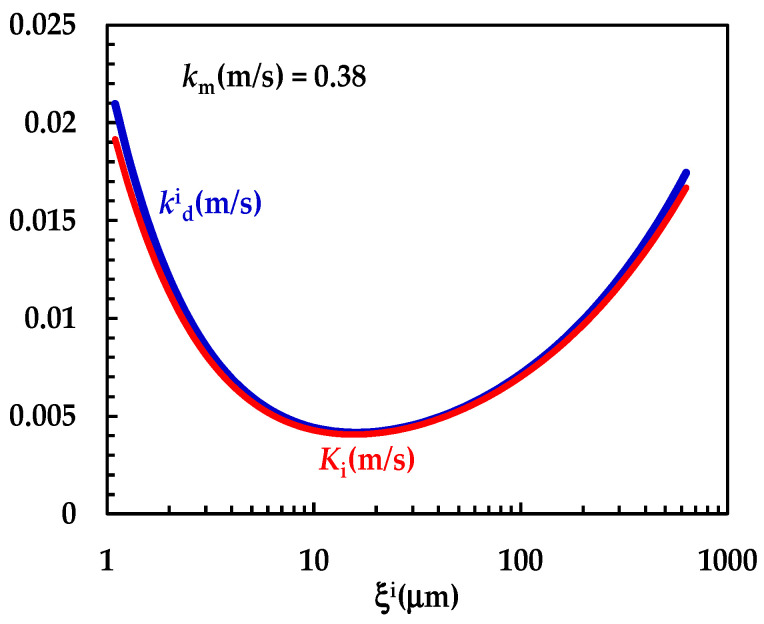
Comparison between *k*_m_ and the $k_{d}^{i}$ − *K*_i_ trend referring to the interpretation of DRT data (TPH) according to the proposed mathematical model.

**Figure 6 pharmaceutics-16-01335-f006:**
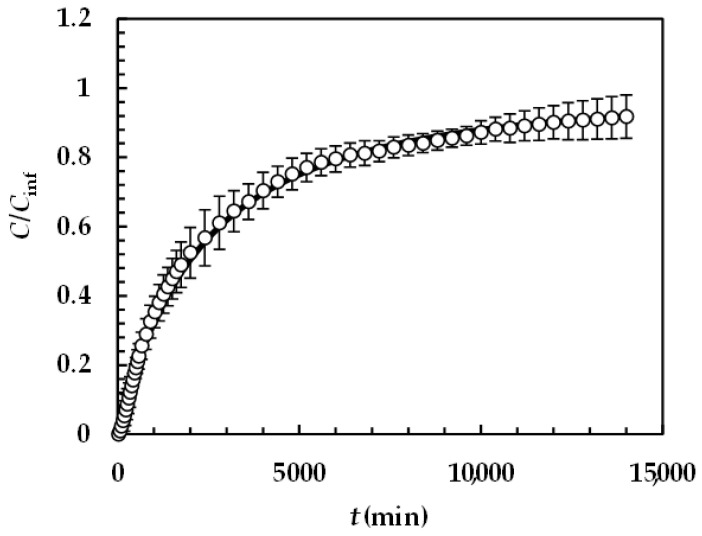
Model’s best fitting (solid line) to experimental DRT data (symbols) referring to PRQ (37 °C). Vertical bars indicate data standard error. The physical parameters adopted to perform data fitting read as follows: ρ_s_ = 1232 kg/m^3^, ρ_f_ = 993 kg/m^3^, η(Pa s) = 6.91 × 10^−3^, ν_f_(m^2^/s) = 0.696 × 10^−6^, *D*(m^2^/s) = 1.0 × 10^−9^, and *C*_s_ (kg/m^3^) = 0.18 [35] while *k*_r_ = *k*_rb_ = 0 as PRQ does not undergo re-crystallization upon dissolution. PRQ particle size distribution is described by the Weibull distribution (Equation (3)), the parameters of which are those reported in Table 4. Concentration data (*C*) are normalized with respect to the final concentration *C*_fin_ = *M*_0_/*V*.

**Figure 7 pharmaceutics-16-01335-f007:**
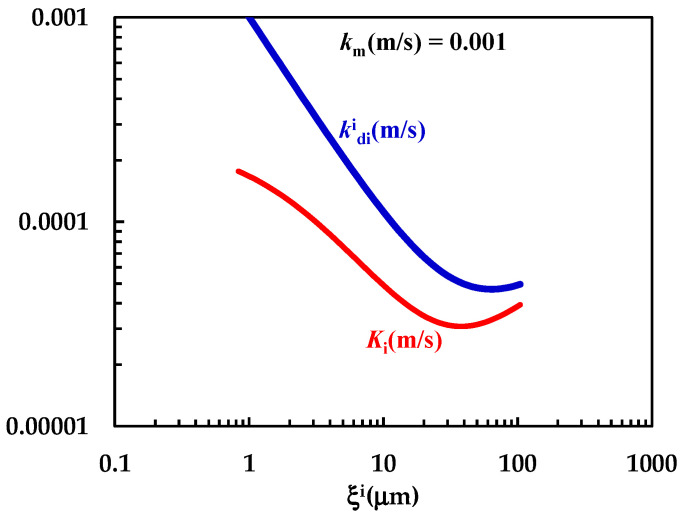
Comparison between *k*_m_ and the $k_{d}^{i}$ − *K*_i_ trend referring to the interpretation of DRT data (PRQ) according to the proposed mathematical model.

**Figure 8 pharmaceutics-16-01335-f008:**
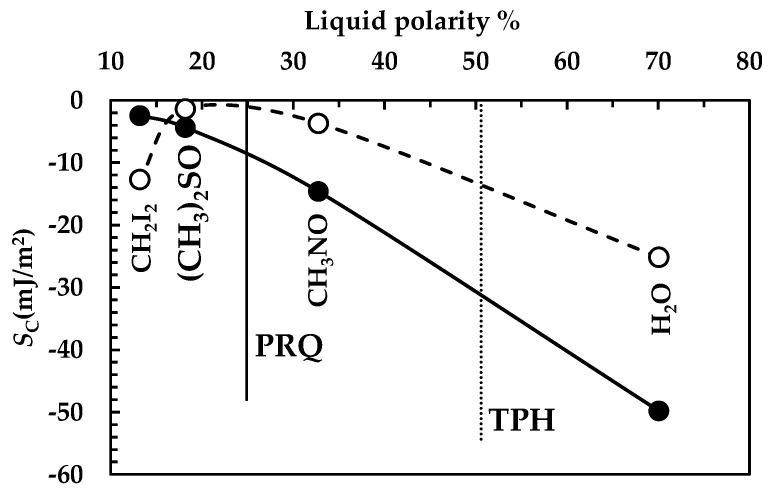
Spreading coefficient (*S*_C_) referring to the four liquids considered (of increasing polarity, see Table 2) on PRQ (solid line) and TPH (dashed line). Vertical solid and dotted lines indicate, respectively, PRQ and TPH’s polarities (see Table 3) (25 °C).

**Table 1 pharmaceutics-16-01335-t001:** Solid–liquid contact angle (θ) referring to TPH and PRQ (25 °C).

Liquid	θ_TPH_	θ_PRQ_
H_2_O	49 ± 2.1	71.6 ± 1.3
CH_3_NO	20.5 ± 4.1	41.6 ± 7.6
(CH_3_)_2_SO	14.4 ± 2.2	25.7 ± 3.0
CH_2_I_2_	41.4 ± 1.2	18.0 ± 2.3

**Table 2 pharmaceutics-16-01335-t002:** Liquid–vapor surface energy (γ_lv_) and relative polar (**γ_lv_^p^**) and dispersion (**γ_lv_^d^**) components (25 °C) [18].

Liquid	γ_lv_ (mJ/m^2^)	γ_lv_^p^ (mJ/m^2^)	γ_lv_^d^ (mJ/m^2^)	Polarity%
H_2_O	71.8	51.0	21.8	70.0
CH_3_NO	68.0	19.0	39.0	32.7
(CH_3_)_2_SO	44.0	8.0	36.0	18.2
CH_2_I_2_	50.8	6.7	44.1	13.2

**Table 3 pharmaceutics-16-01335-t003:** Solid–vapor surface energy (γ_sv_) and relative polar (**γ_lv_^p^**) and dispersion (**γ_lv_^d^**) components evaluated according to the Wu approach.

Drug	γ_sv_ (mJ/m^2^)	γ_sv_^p^ (mJ/m^2^)	γ_sv_^d^ (mJ/m^2^)	Polarity%
Theophylline	55.1	27.8	27.2	50.6
Praziquantel	50.6	38.0	12.6	24.9

**Table 4 pharmaceutics-16-01335-t004:** Fitting parameters of the Weibull distribution (Equation (3)) referring to the two model drugs considered. *R*_A_ is the average radius of the distribution.

Drug	*R*_min_ (μm)	*ϕ* (μm)	*φ* (-)	*R*_A_ (μm)
Theophylline	0.48	218.9	1.266	109.6
Praziquantel	0.79	35.3	1.256	18.5

## Data Availability

All the presented data are available upon request to the corresponding author.

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
