# Peer review of "The Key Role of Wettability and Boundary Layer in Dissolution Rate Test"

_pharmaceutics, 2024, doi:10.3390/pharmaceutics16101335_

Round 1

Reviewer 1 Report

Comments and Suggestions for Authors

The proposed mathematical model for describing the dissolution of poly-disperse drug spherical particle solution (DRT) is a comprehensive and well-structured approach to understanding drug bioavailability in the pharmaceutical field. The model considers key hallmarks of DRT, including particle size distribution, solubility, wettability, and hydrodynamic conditions, and applies the spherical shape simplification to two model drugs, theophylline and praziquantel.

The results of the data analysis within the proposed model provide valuable insights into the dissolution behavior of the two model drugs, highlighting the importance of factors such as boundary layer resistance and wettability. Valuing results by determining drug wettability and polarity adds credibility to the proposed model. The successful fitting of the DRT data for theophylline and praziquantel using just two fitting parameters demonstrates the robustness and applicability of the model.

Overall, the proposed mathematical model effectively captures the different physical phenomena governing the dissolution of poly-disperse solid drug particle solution and provides valuable insights for assessing drug bioavailability.  I don't feel any modification is required in the current manuscript, which is presented well; therefore, I suggest accepting it in its current form. 

Author Response

Referee 1

1) The proposed mathematical model for describing the dissolution of poly-disperse drug spherical particle solution (DRT) is a comprehensive and well-structured approach to understanding drug bioavailability in the pharmaceutical field. The model considers key hallmarks of DRT, including particle size distribution, solubility, wettability, and hydrodynamic conditions, and applies the spherical shape simplification to two model drugs, theophylline and praziquantel.

We thank the referee for this comment.

2) The results of the data analysis within the proposed model provide valuable insights into the dissolution behavior of the two model drugs, highlighting the importance of factors such as boundary layer resistance and wettability. Valuing results by determining drug wettability and polarity adds credibility to the proposed model. The successful fitting of the DRT data for theophylline and praziquantel using just two fitting parameters demonstrates the robustness and applicability of the model.

We thank again the referee for this comment.

3) Overall, the proposed mathematical model effectively captures the different physical phenomena governing the dissolution of poly-disperse solid drug particle solution and provides valuable insights for assessing drug bioavailability.  I don't feel any modification is required in the current manuscript, which is presented well; therefore, I suggest accepting it in its current form.

We are delighted that the referee appreciated our approach.

Reviewer 2 Report

Comments and Suggestions for Authors

The manuscript provides an insight into the wettability of solid surface in regard to dissolution. Multiple solvents with different polarities were used to test the contact angle and the spreading coefficient. The dissolution testing was performed only in the water. Do you have dissolution data in other solvents to share? The conclusion doesn't match with the study. Revise the conclusion. 

Author Response

Referee 2

1) The manuscript provides an insight into the wettability of solid surface in regard to dissolution. Multiple solvents with different polarities were used to test the contact angle and the spreading coefficient. The dissolution testing was performed only in the water.

Do you have dissolution data in other solvents to share?

We thank the reviewer for the interest and attention to our work. Herein, we did not present data in other solvents, as the core target of the study relied on mimicking the in-vivo dissolution (DRT) behavior. Nevertheless, the developed mathematical model can be properly applied to the DRT process occurring in other solvents, too. Indeed, as we hypothesized on the “Introduction” section, the hydrophilic or hydrophobic nature of the solvent does not affect the model robustness, according to the Results recorded.

2) The conclusion doesn't match with the study. Revise the conclusion.

We thank the reviewer for the useful comment, which allow us to improve the manuscript. Hence, the “Conclusions” section has been revised (see red text).

Reviewer 3 Report

Comments and Suggestions for Authors

The key role of wettability and boundary layer on Dissolution Rate Test

This manuscript contains the original research work in which the authors propose a mathematical model able to describe the dissolution of poly-disperse drug spherical particles solution (Dissolution Rate Test - DRT). DRT is a pivotal test performed in the pharmaceutical field to qualitatively assess drug bioavailability. Two model drugs were considered to assess the robustness of model: theophylline (soluble and wettable) and praziquantel (poorly soluble but wettable).The analysis of the obtained results suggests that the proposed mathematical model confirms the importance of the different physical phenomena leading the dissolution of poly-disperse solid drugs particles solution. Although a comprehensive mathematical model was proposed and applied, the DRT data of theophylline and praziquantel was successfully fitted by means of just two fitting parameters.

I have no fundamental remarks on the work. All the obtained results are carefully analyzed, and the conclusions drawn are justified. I recommend accepting the current manuscript, however there are some aspects in this manuscript that should be improved:

Comment 1: I suggest the authors rewrite the last paragraph of the Introduction section to clearly state the objectives, mention the hypothesis, and highlight the significance of the study.

Comment 2: Please unify the font color throughout the manuscript, for example, in lines 31, 75, etc. Also, when citing figure and table references.

Comment 3: Please explain the abbreviation COPD.

Comment 4: DRT test: Please include the calibration curve of the UV spectrophotometer method and the concentration range for TPH and PRQ. Line 176: Explain in the text why using 3 mg TPH and 27 mg PRQ.

Comment 5: Page 10, line 378: Explain in the text why TPH monohydrate does not undergo re-crystallization during dissolution process.

Comment 6: Page 11, line 407: Explain in the text why PRQ does not undergo re-crystallization with a monotonic trend.

Comment 7: Conclusions: please use a single paragraph, include the two parameters mentioned in line 470 and expand the conclusions to emphasize the importance of the work and strengthen it.

Author Response

Referee 3

1) This manuscript contains the original research work in which the authors propose a mathematical model able to describe the dissolution of poly-disperse drug spherical particles solution (Dissolution Rate Test - DRT). DRT is a pivotal test performed in the pharmaceutical field to qualitatively assess drug bioavailability. Two model drugs were considered to assess the robustness of model: theophylline (soluble and wettable) and praziquantel (poorly soluble but wettable). The analysis of the obtained results suggests that the proposed mathematical model confirms the importance of the different physical phenomena leading the dissolution of poly-disperse solid drugs particles solution. Although a comprehensive mathematical model was proposed and applied, the DRT data of theophylline and praziquantel was successfully fitted by means of just two fitting parameters. I have no fundamental remarks on the work. All the obtained results are carefully analyzed, and the conclusions drawn are justified. I recommend accepting the current manuscript, however there are some aspects in this manuscript that should be improved:

We would like to thank the referee for the comments.

2) Comment 1: I suggest the authors rewrite the last paragraph of the Introduction section to clearly state the objectives, mention the hypothesis, and highlight the significance of the study.

The last paragraph of the “Introduction” section has been revised, according to this comment (please see red text within the revised manuscript version).

3) Comment 2: Please unify the font color throughout the manuscript, for example, in lines 31, 75, etc. Also, when citing figure and table references.

We agree and thank the reviewer for this comment. The font color has been unified throughout the paper.

4) Comment 3: Please explain the abbreviation COPD.

We thank the reviewer for the fruitful note. Therefore, the abbreviation related to what COPD stands for, has been introduced in the revised manuscript version (see red text).

5A) Comment 4: DRT test: Please include the calibration curve of the UV spectrophotometer method and the concentration range for TPH and PRQ.

We thank the reviewer for the suggestion. Thus, the calibration curve and the concentration range have been added in the revised manuscript version, Appendix A.

5B) Line 176: Explain in the text why using 3 mg TPH and 27 mg PRQ.

We thank the reviewer for the suggestion. First of all, PRQ is remarkably less soluble than TPH and it is characterized by a smaller extinction coefficient, compared to the TPH’s one. Therefore, it was required to strongly increase the PRQ amount (with reference to TPH), to be able to follow and record the beginning part of the DRT curve, too. For the sake of conciseness, we skipped the description of the long tuning process required to get an optimal experimental set up. Overall, thanks to the fruitful reviewer’s insight, we provided to add on this detailed explanation in the revised text (see red text), to increase the manuscript quality and understanding.

6) Comment 5: Page 10, line 378: Explain in the text why TPH monohydrate does not undergo re-crystallization during dissolution process.

We thank the reviewer for the suggestion. Basically, organic drugs can undergo re-crystallization, when their structure thermodynamic equilibrium can be altered by the presence of the solvent. Thus, during the dissolution process, an amorphous drug can usually transform into its more stable crystalline form. On the other hand, the solvent can induce a polymorphic transition, when the drug can exist in different polymorphic forms. Moreover, in case of solvated forms, the anhydrous form transforms into its proper, more stable solvated one, in aqueous dissolution medium. All the mentioned transformations lead to a reduction of drug solubility [ref 18, over the manuscript file]. As we used throughout this study the hydrated form of TPH, it did not undergo any re-crystallization phenomena, which can be expected for its anhydrous form. Overall, thanks to the fruitful reviewer’s insight, we provided to add on this detailed explanation in the revised text (see red text), to increase the manuscript quality and understanding.

7) Comment 6: Page 11, line 407: Explain in the text why PRQ does not undergo re-crystallization with a monotonic trend.

We thank the reviewer for the suggestion, which matches comment 5. As water does not induce in PRQ any of the three transformations mentioned in our answer on comment 5, this drug does not undergo re-crystallization. Overall, thanks to the fruitful reviewer’s insight, we provided to add on this detailed explanation in the revised text (see red text), to increase the manuscript quality and understanding.

8) Comment 7: Conclusions: please use a single paragraph, include the two parameters mentioned in line 470 and expand the conclusions to emphasize the importance of the work and strengthen it.

We thank the reviewer for the suggestion, hence, the “Conclusions” section has been revised accordingly (see red text).

Round 2

Reviewer 2 Report

Comments and Suggestions for Authors

The author has complied with the suggestions